# Regional Transmission and Reassortment of 2.3.4.4b Highly Pathogenic Avian Influenza (HPAI) Viruses in Bulgarian Poultry 2017/18

**DOI:** 10.3390/v12060605

**Published:** 2020-06-01

**Authors:** Divya Venkatesh, Adam Brouwer, Gabriela Goujgoulova, Richard Ellis, James Seekings, Ian H. Brown, Nicola S. Lewis

**Affiliations:** 1Department of Pathobiology and Population Sciences, Royal Veterinary College, Hatfield, Hertfordshire AL9 7TA, UK; nilewis@rvc.ac.uk; 2OIE/FAO/ International Reference Laboratory for avian influenza, swine influenza and Newcastle Disease, Animal and Plant Health Agency (APHA), Weybridge, Addlestone, Surrey KT15 3NB, UK; adam.brouwer@apha.gov.uk (A.B.); james.seekings@apha.gov.uk (J.S.); ian.brown@apha.gov.uk (I.H.B.); 3National Diagnostic Research Veterinary Medical Institute, 1231 Sofia, Bulgaria; gvgoujgoulova@abv.bg; 4Surveillance and Laboratory Services Department, Animal and Plant Health Agency (APHA), Weybridge, Addlestone, Surrey KT15 3NB, UK; Richard.Ellis@apha.gov.uk; 5Virology Department, Animal and Plant Health Agency (APHA), Weybridge, Addlestone, Surrey KT15 3NB, UK

**Keywords:** avian influenza, HPAI (highly pathogenic avian influenza), 2.3.4.4b, poultry outbreak, genomic epidemiology

## Abstract

Between 2017 and 2018, several farms across Bulgaria reported outbreaks of H5 highly-pathogenic avian influenza (HPAI) viruses. In this study we used genomic and traditional epidemiological analyses to trace the origin and subsequent spread of these outbreaks within Bulgaria. Both methods indicate two separate incursions, one restricted to the northeastern region of Dobrich, and another largely restricted to Central and Eastern Bulgaria including places such as Plovdiv, Sliven and Stara Zagora, as well as one virus from the Western region of Vidin. Both outbreaks likely originate from different European 2.3.4.4b virus ancestors circulating in 2017. The viruses were likely introduced by wild birds or poultry trade links in 2017 and have continued to circulate, but due to lack of contemporaneous sampling and sequences from wild bird viruses in Bulgaria, the precise route and timing of introduction cannot be determined. Analysis of whole genomes indicates a complete lack of reassortment in all segments but the matrix protein gene (MP), which presents as multiple smaller clusters associated with different European 2.3.4.4b viruses. Ancestral reconstruction of host states of the hemagglutinin (HA) gene of viruses involved in the outbreaks suggests that transmission is driven by domestic ducks into galliform poultry. Thus, according to present evidence, we suggest the surveillance of domestic ducks as they are an epidemiologically relevant species for subclinical infection. Monitoring the spread due to movement between farms within regions and links to poultry production systems in European countries can help to predict and prevent future outbreaks. The 2.3.4.4b lineage which caused the largest recorded poultry epidemic in Europe continues to circulate, and the risk of further transmission by wild birds during migration remains.

## 1. Introduction

Aquatic birds form a reservoir for avian influenza viruses, where multiple subtypes circulate and generally do not cause any disease. Periodically, such viruses can infect gallinaceous poultry as low pathogenic avian influenza (LPAI) viruses. Here, viruses of H5 and H7 hemagglutinin (HA) subtypes can mutate into highly pathogenic forms [1,2,3,4]. Highly pathogenic avian influenza (HPAI) viruses constitute a major threat to poultry populations worldwide because of their high levels of mortality, potential for spread, impact on livestock production and their zoonotic potential.

Such LPAI to HPAI transformations have occurred on multiple occasions via an introduction of multiple basic amino acids in the HA proteolytic cleavage site. After influenza viruses enter the cell, there is a cleavage of the inactive precursor HA0 protein at this site into HA1 and HA2 subunits. The regular cleavage site is cleaved by trypsin or trypsin-like proteases present largely in the respiratory and intestinal epithelia. However, the multi-basic cleavage site can be cleaved by several ubiquitous proteases such that the virus can infect and replicate in multiple tissue types resulting in a systemic disease with high morbidity and mortality [5,6]. 

Outbreaks caused by certain HPAIs of the A/goose/Guangdong/1/1996 (Gs/GD) lineage of H5 viruses continued to circulate in domestic birds in some countries and were re-introduced into wild aquatic bird reservoirs to spread to new geographic areas [7,8]. Thus far, viruses bearing this Gs/GD lineage HA have caused infections in poultry, wild birds and humans in up to 83 countries across several continents (EMPRES-i data http://empres-i.fao.org/eipws3g/ accessed 9 April 2019, [9]). Continued circulation of viruses bearing this H5 HA gene in poultry and wild aquatic reservoirs has resulted in its diversification into multiple clades and sub-clades; by 2012, 12 distinct clades were identified [10]. 

Viruses bearing HAs of the Gs/GD 2.3.4 lineage emerged around 2009–2013 and revealed an early propensity to reassort with neuraminidase (NA) subtypes other than N1, unlike earlier clades, and showed unprecedented geographical range expansion via the poultry trade and wild bird migration [11,12,13]. Since 2014, HPAI clade 2.3.4.4 viruses have spread rapidly through Eurasia and into North America via migratory wild aquatic birds and have evolved through a reassortment with prevailing local low pathogenicity avian influenza viruses. They are associated with variable disease severity, including subclinical infection in wild birds and domestic waterfowl [14]. From May 2016, the clade 2.3.4.4 Group B (2.3.4.4b) H5N8 viruses re-emerged in Europe, causing numerous outbreaks in poultry and a large number of deaths in wild birds [15,16,17,18]. Several reassortment events led to the emergence and detection of HPAI H5N5 in several European countries, Georgia and Israel between November 2016 and June 2017 [19] and HPAI H5N6 in Greece in February 2017 [20], with the subsequent wave of HPAI H5N6 viruses evolving from the H5N8 2016–2017 viruses during 2017 by reassortment of a European HPAI H5N8 virus and a wild host reservoir of LPAI viruses [18,21]. 

With over 1197 H5 HPAI outbreaks reported in poultry or captive birds in 20 countries, the 2016/2017 HPAI epidemic was the largest ever recorded in the EU in terms of the number of outbreaks, geographic distribution and the number of dead wild birds [19,22]. Reports indicate that the H5N8 virus persisted during early winter of 2016 into the late summer of 2017 at least, leading to sporadic outbreaks in poultry and wild bird infections into the autumn [22]. Events decreased in early winter 2017 with 48 poultry outbreaks and nine wild bird detections recorded towards the end of the winter of 2017, with even fewer outbreaks and detections in wild birds the following year [22,23]. 

Viruses of the H5 subtype were largely absent through 2018 and past February until December 2019 only, when two wild bird HPAI events were identified both in Denmark, but Bulgaria remained the only country in Europe reporting HPAI outbreaks in poultry holdings [24,25]. In this study we use genomic and traditional epidemiological methods to source and track the spread of HPAI poultry outbreaks that occurred in Bulgaria during 2017/18. We aim to understand the role of wild birds and the poultry system structure in HPAI introduction, onward spread in poultry and the subsequent potential for endemic circulation of highly pathogenic viruses in domestic birds to help design better containment and preventive measures. 

## 2. Materials and Methods

### 2.1. Epidemiology

Epidemiological data on infected premises were obtained from the Animal Disease Notification System managed by the European Commission and validated with the National Diagnostic Research Veterinary Medical Institute in Bulgaria. Isolates obtained from the veterinary authorities in Bulgaria were matched to the appropriate infected premise and GIS analyses was performed using ArcGIS Desktop 10.2.2 [26].

Additional background data were also taken from Bulgarian presentations to the Standing Committee on Plants, Animals, Food and Feed [27].

### 2.2. Sequencing

RNA was sequenced using the MiSeq platform. Briefly, viral RNA was extracted from the nasal swab using the QIAmp viral RNA mini kit without the addition of carrier RNA (Qiagen, Manchester, UK). cDNA was synthesised from RNA using a random hexamer primer mix and cDNA Synthesis System (Roche, UK). The sequence library was prepared using a NexteraXT kit (Illumina, Cambridge, UK). Quality control and quantification of the cDNA and the sequence library was performed using Quantifluor dsDNA System (Promega, Southampton, UK). Sequence libraries were run on a Miseq using MiSeq V2 300 cycle kit (Illumina, Cambridge, UK) with 2 × 150 base paired end reads. The raw sequence reads were analysed using publicly available bioinformatics software, following an in-house pipeline, available on github (https://github.com/ellisrichardj/FluSeqID/blob/master/FluSeqID.sh). This pipeline de novo assembles the raw data using the Velvet assembler [28] and Basic Local Alignment Search Tools (BLASTs) and the resulting contigs against a local database of influenza genes using Blast+ [29] and then maps the raw data against the highest scoring blast hit using the Burrows-Wheeler Aligner [30]. The consensus sequence was extracted from the resultant bam file using a modified SAMtools software package [31], script (vcf2consensus.pl) available at: https://github.com/ellisrichardj/csu_scripts/blob/master/vcf2consensus.pl. Whole-genome sequences of the viruses are available on GISAID database with unique IDs: EPI_ISL_419212, EPI_ISL_419344, EPI_ISL_419345, EPI_ISL_419346, EPI_ISL_419347, EPI_ISL_419348, EPI_ISL_419349, EPI_ISL_419350, EPI_ISL_419351, EPI_ISL_419352, EPI_ISL_419353, EPI_ISL_419354, EPI_ISL_419355, EPI_ISL_419356, EPI_ISL_419357, EPI_ISL_419358, EPI_ISL_419359, EPI_ISL_419360, EPI_ISL_419361, EPI_ISL_419362, EPI_ISL_419363, EPI_ISL_419364, EPI_ISL_419365, EPI_ISL_419366, EPI_ISL_419367.

### 2.3. Visualising Reassortment

We first used BLAST on GISAID [32,33] to identify the closest genetic relatives to the Bulgarian outbreak viruses. We found that for all segments, the top 50 blast hits were largely from viruses bearing H5 HA and were isolated from avian hosts in Europe in the year 2016 or later. Therefore, for the reassortment analysis, we downloaded whole-genome sequences from all H5Nx viruses isolated between January 2016 and December 2018 from avian hosts in Europe from FluDB and GISAID. Sequences from each segment were checked for quality including sequence length > 30% average length for a given segment, removing duplicates, and all 8 segments present. The resulting datasets were combined with Bulgarian segment sequences. 

Final datasets were aligned using MAFFT v7.305b [34] and trimmed to only retain nucleotides from the starting ATG until the final STOP codon. We inferred Maximum Likelihood (ML) phylogenetic trees for each gene segment using IQ-TREE, 1.5.5 [35] and obtained branch supports with Shimodaira-Hasegawa (SH)-like approximate Likelihood Ratio Test (aLRT, 1000 replicates). 

BALTIC (backronymed adaptable lightweight tree import code, https://github.com/evogytis/baltic) was used to compare the phylogenetic structure of the segment genes. The phylogenetic position of each strain was traced and coloured according to the lineage and location across unrooted ML trees for HA and all internal gene segments. Figures were generated by modifying scripts from a similar analysis [36] and were edited in Adobe Illustrator. We selected a qualitative palette of colors using http://colorbrewer2.org/.

### 2.4. Spatial Phylodynamic Analysis and Ancestral Host Reconstruction

For the H5 analysis, 1189 HA sequences of strains isolated between January 2016 and December 2018 were downloaded from GISAID (Appendix A). 

The individual H5 dataset was first subjected to a quality control step where all duplicate sequences and sequences bearing duplicate IDs were removed (where 871 sequences remained). These sequences, together with those sequenced from Bulgaria (BGR), were aligned using MAFFT v7.305b [34] and used in the FastTree program [37] to generate a maximum-likelihood tree with a GTR+gamma substitution model. All sequences from taxa outside of the fully supported (100%) cluster with BGR were discarded. The H5 HA dataset included representatives from the HPAI clade 2.3.4.4b (H5N8) 2016-17 clade (need WHO H5 group nomenclature reference) to form a new dataset of 104 sequences from which a final H5 ML tree was inferred using FastTree. This tree was analyzed with tempest v1.5 to check for clock-like behaviour [38]. 

Bayesian phylogenetic trees were inferred using BEAST v1.10.4 [39] to determine the time of the emergence of the Bulgarian viruses. We coded each taxon with host (anseriformes, galliformes, other) and location (Dobrich, Haskovo, Plovdiv, Sliven, Stara Zagora, Yambol) information. Akaike’s information criterion through Markov chain Monte Carlo (AICM) values were used to select the appropriate state (symmetric vs asymmetric) and clock (strict vs relaxed (uncorrelated relaxed lognormal)) models. Asymmetric state transition and strict clock were chosen over symmetric/relaxed clock. Gaussian Markov random field (GMRF) Bayesian Skyride population prior was used with a random starting tree. All other priors were set to default. Markov chain monte carlo (MCMC) was set to 50,000,000 generations. Two separate runs were performed to ensure convergence between runs. Log files were analysed in Tracer v1.7.1 to determine convergence, and to check that ESS values were beyond the threshold (>200). Log and trees files from both runs were combined using Log Combiner v 1.10.4. Tree annotator v1.10.4 was used to generate a maximum credibility tree (MCC) using 10% burn in and median node heights. The MCC tree was then annotated to include posterior probability values and time scales, and host and location states, and were plotted in R v 3.5 using the ggtree package [40].

## 3. Results

### 3.1. Epidemiology

Between 17 October 2017 and 8 April 2019, Bulgaria reported 38 HPAI H5 outbreaks in poultry. Bulgaria accounted for 43.6% of outbreaks in Europe in this timeframe. During this period there were no positive wild birds detected in Bulgaria among the 47 and 58 dead or moribund wild birds sampled through passive surveillance in 2017 and 2018 respectively [41]. Bulgaria did not undertake any active wild bird surveillance during the study period. As Figure 1 shows, we observed four loose geographical clusters in the infected premise locations: Plovdiv and Haskovo in Central Bulgaria, Yambol in East Bulgaria and Dobrich in North-East Bulgaria and sporadic cases in Vidin and Lovech in North-West Bulgaria. A range of species were affected including ducks (19), chickens (7), turkeys (1) partridges (1) and backyard poultry including hens and ducks (10). There was no specific-species bias between clusters. Both backyard and commercial settings were affected.

Clinical indicators especially including mortality typical of HPAI were sufficient triggers for vet investigation of premises. Mortality in Anseriformes birds was variable with a number of early events showing mortality, but later events showing little or no mortality. 

Of the 38 infected premises, samples from 25 locations were sent to Animal and Plant Health Agency (APHA), UK for confirmatory testing, virus isolation and genetic analyses. Samples from infected premises detected after 31 October 2018 were not sent for confirmatory testing, and APHA did not receive samples from two outbreaks in ducks confirmed on the 10 April 2018 and 24 May 2018 in Plovdiv (Submission ID 2018/3 and 2018/8). 

As Figure 2 shows, commercial duck production in Bulgaria is concentrated in the south/central and eastern parts of the country with the north and the west having relatively few registered holdings, whilst chicken holdings are more homogenously located throughout the country. Virologically positive poultry-infected premises were found primarily in the high duck density areas of Bulgaria, for both duck outbreaks as well as gallinaceous poultry outbreaks.

### 3.2. Genetic Structure of Bulgarian H5 HA

The maximum-likelihood tree in Figure 3 shows that the highly pathogenic H5 HAs from Bulgaria arose from European H5N8 strains of the 2.3.4.4b lineage. They show no close genetic link with the subsequent H5N6 wave from Asia that spread into Europe [18,21]. Bulgarian virus HAs isolated in 2017–18 form two separate clusters originating from distinct 2.3.4.4b H5 virus ancestors, together with a lone 2017 strain from Dobrich. The clusters are structured geographically with strains exclusively from Dobrich forming one cluster in the northeast (in green). Strains from Central Bulgarian regions Plovdiv, Haskovo and Stara Zagora (in red), eastern regions such as Yambol and Sliven (in blue), and Vidin in the west (in purple) form a single mixed cluster (see Appendix A for representation on a map). This implies that viruses in both clusters are circulating separately and their HAs likely have separate origins. Within both clusters, sub-clusters with possible separate origins are discernible but the data is likely confounded by under-sampling.

### 3.3. Timing of Introductions, Host Dynamics and Spatial Spread

The BEAST maximum clade credibility (MCC) tree for the HA gene (Figure 4A) shows that after introduction, virus HAs in the mixed cluster have been circulating within Bulgaria since approximately early 2017 (95% highest posterior density interval (HPD): September 2016–May 2017) and those in the Dobrich cluster, since May 2017 (95% HPD: March–October 2017). This means that after the two separate introductions into Bulgarian poultry, the viruses have been transmitted locally within the country. The lone October 2017 Dobrich virus HA was a separate introduction which failed to spread.

Location-wise, it is likely that the source of the mixed cluster of viruses was Plovdiv, whence it spread to other regions in Central and Eastern Bulgaria (Figure 4B and Figure 5). While the monophyly of the Bulgarian mixed and Dobrich clades are very well-supported (posterior probability (pp) = 0.999), the identity of the closest related virus is unclear for both the mixed and Dobrich clusters (pp = 0.046, 0.478) which is likely due to a lack of sampling. Even though 2 and 14 HPAI H5 viruses were detected in wild birds in Bulgaria in 2016 and 2017 respectively, we do not have sequences from these viruses to determine how they relate to the poultry outbreaks. No onward transmission to other geographic areas were detected as of early 2019. Although there is evidence of serologically positive poultry within Bulgaria in 2018–19, we were unable to explore their provenance due to a lack of viruses or genetic sequences. 

Within poultry, the transmission occurred largely in a direction away from ducks (anseriformes) to chickens (galliformes), but not from chickens back into ducks (Figure 6). This pattern is consistent with our epidemiological findings above as well as previous studies in wild birds [42] and other reported 2.3.4.4b poultry outbreaks in Europe which implicated ducks in the local spread [43,44].

### 3.4. Whole-Genome Analysis

We used BALTIC (backronymed adaptable lightweight tree import code, https://github.com/evogytis/baltic) to compare the phylogenetic structure of the internal genes of the Bulgarian strains compared to other HPAI H5 and LPAI viruses. To visualize incongruence, the phylogenetic position of each sequence (coloured according to the origin of its HA) was traced across all eight trees (Figure 7). Parallel lines indicate a similar origin, whereas crossed lines indicate differential ancestry. We find that the internal genes of the Bulgarian strains are largely derived from the same ancestral viruses, the lines connecting the strains are largely parallel and remain within the same larger cluster of H5 viruses. Some variation is present for the MP gene, which shows multiple clusters with each one related to MP segments that are associated in turn with other European 2.3.4.4b viruses isolated from hosts such as ducks, mute swans and pheasants (Appendix A); if there are any Bulgarian intermediaries, they remain unsampled Like the HAs, none of the segments appear to be related to any viruses from the recent 2.3.4.4b H5N6 wave in Asia that spread into Europe, but to the dominant 2.3.4.4b H5N8 lineage. 

This pattern of limited reassortment is consistent with virus transmission and endemic maintenance within Bulgarian poultry and shows no epidemiological role for currently circulating low pathogenic avian influenza in wild birds. 

## 4. Discussion

In this study we investigated an apparently unique and independent epidemiological event in Bulgaria in 2017–18. During this study period and through to 2019 there was no evidence of sustained transmission or endemicity of other H5N8 2.3.4.4b virus outbreaks in Europe: EU co-funded mandatory surveillance for Avian Influenza in the EU consists of a serosurveillance programme for notifiable subtypes which will provide evidence of exposure within a population if there has been non-negligible spread between flocks [41]. Results from the EU active serological surveillance programme in Bulgaria for poultry show no positive serological results between 2016 and 2018, despite sampling around 500 poultry holdings a year of which 109, 276 and 155 duck holdings were sampled respectively in 2016, 2017 and 2018 (Appendix A). There was also a compulsory programme of passive surveillance in dead or moribund wild birds that is co-financed by the EU (as stated by European Union guidelines [45]). Results from this surveillance for 2017 and 2018 are shown in Appendix A alongside results from the two EU member-states of Romania and Greece that share a land border with Bulgaria. Whilst there were initial detections of H5N8 HPAI in southeast Europe in late 2016, these continued and accelerated in early 2017 with the last confirmed H5N8 case in wild birds found in Romania in March 2017. There were no identified cases of H5N8 found in wild birds in the region throughout the remainder of 2017 and no cases identified in 2018.

We were able to trace the Bulgarian outbreaks during 2017–18 to the 2.3.4.4b lineage viruses circulating in Europe in 2017. As in a previous Bulgarian study [43], we found no links to viruses from Bulgarian wild birds despite concurrent sampling in the earlier study; nor did we find links to Eurasian/East Asian wild birds. We demonstrated that incursion of these viruses into Bulgarian domestic birds has occurred on at least two occasions into different regions of the country, from distinguishable genetically distinct 2.3.4.4b ancestral strains which, after introduction, have circulated with spatial separation within the country since 2017. Phylogenetically long branch lengths connecting to poultry viruses also indicate significant in-country circulation of undetected disease and unsampled strains for a period before outbreaks. From available data we cannot say if the route of transmission was via migratory wild birds or duck trade links between Bulgaria and other European countries. 

During the outbreaks, we show that transmission into galliform poultry is likely driven by domestic ducks. Given the presence of limited reassortment signals (Figure 7) and an absence of detections in wild birds in Bulgaria or any other parts of Europe during 2018, H5N8 viruses have been likely maintained endemically within the domestic bird sector. However, for the MP gene segment (see Appendix A) we did find multiple clusters of viruses, each associated with isolates from multiple species (ducks, swan, pheasant) in different countries (Netherlands, England, Hungary). A wider sampling of domestic ducks might be revealed if the limited reassortment that was detected in the MP gene segment is driven by transmission from these hosts. Additionally, H5N8 viruses identified in 2019 in locations such as Poland and Slovakia appear to be unrelated to Bulgarian viruses [46]. Although we do not have sequences yet, H5 HPAI viruses continue to be detected in Bulgarian poultry (as per information downloaded from http://empres-i.fao.org/eipws3g/, see Appendix A). Once sequences are available, genetic analysis will reveal whether they are related to the viruses we describe in this study, either revealing continued transmission or a new introduction from wild birds. 

We note the risk of the domestic duck sector as a source of cryptic spread and maintenance after introduction. Our results are consistent with detailed spatial modelling during the 2016–17 HPAI 2.3.4.4b outbreaks in France [44,47], and previous surveillance in Bulgaria during 2008–12 [43]. Both studies found a major role for duck production systems, particularly foie gras, in the spread of avian influenza in the poultry sector. France and Bulgaria together with Hungary are the major producers of foie gras in Europe and maintain trade links connecting duck farms (transport of ducklings and eggs) in all three countries [43]. Ducks often do not show clinical signs of disease, which makes identification of infection challenging. In addition, short production cycles, high movement of personnel and ducks between farms and challenges in cleaning and disinfection of transport vehicles also contribute to the potential maintenance of influenza A viruses without detection [43,47,48,49]. Serological and swab testing of ducks as sentinel species might be useful in predicting future outbreaks in poultry. 

Our analysis demonstrates movement of viruses in Bulgaria between farms, but the viruses are largely regionally sequestered. There is no evidence for movement between regions, for example, between Dobrich in the northeast and Plovdiv in Central Bulgaria. However, connections between Plovdiv and Eastern Bulgaria (Yambol, Sliven) which is geographically relatively close are clear. Vigilance should also be maintained for indirect transmission potential, as a more recently isolated virus from the western region of Vidin appears to be closely related to viruses found in Central Bulgaria, indicating a possible link between these regions either via wild birds or translocation on land. 

The study has demonstrated the challenge of the relatively low biosecurity barrier between the wild bird interface and domestic duck production in Bulgaria. The nature of the farming presents many challenges for strong bio security measures that may be applicable in other production sectors. However, the evidence in this study shows the risk of lateral spread within the duck sector in particular. This information should be informative to industry stakeholders who should review practices for functional connection and activity relating to husbandry within the duck sector. This in turn should facilitate the strengthening of biosecurity through introduction of phytosanitary practices that would have the effect of breaking the chain of infection, such as thorough disinfection of equipment moving between different sites. The frequency of movements as part of the husbandry cycle can be problematical and greater rigor in interventions designed to apply rigorous sanitation measures at key production cycle points should be considered. Future studies to understand the spread of HPAI viruses and design improved interventions should focus on sampling domestic ducks to establish their role in virus maintenance and transmission, examine regional movement of people and commodities including equipment/vehicles between poultry farms within Bulgaria, and investigate trade links with poultry production systems in other European countries (especially Hungary and France) along with evaluating targeted production-system specific approaches to avian influenza surveillance and control.

Thus far, even though no onward transmission of these viruses from Bulgaria to other geographic regions have been detected, there is serological evidence of continued infections within Bulgarian poultry. Therefore, the H5 2.3.4.4b lineage which caused the largest recorded poultry epidemic in Europe continues to circulate and a risk of both further outbreaks in poultry and also a spill-over back into wild birds and further dispersal during migration remains.

## Figures and Tables

**Figure 1 viruses-12-00605-f001:**
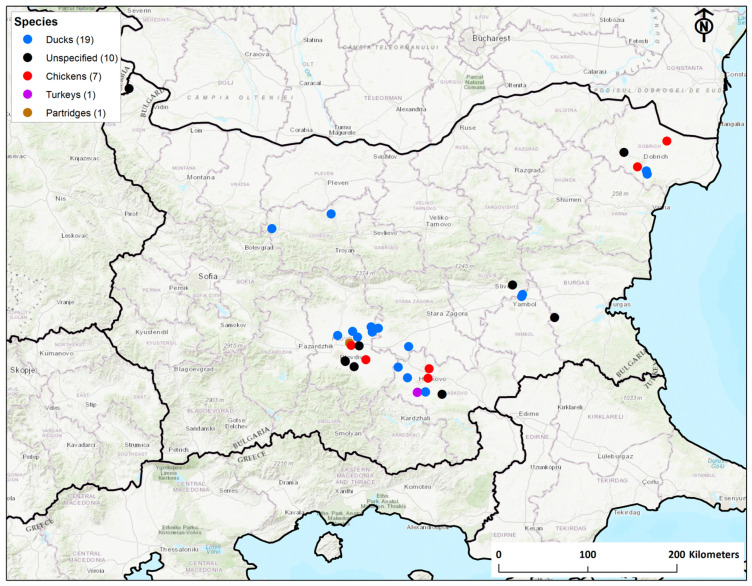
H5N8 highly pathogenic avian influenza (HPAI) outbreaks in poultry in Bulgaria 2017–2019. Color of dots indicates species affected as shown in the key. Numbers in brackets indicate the number of premises in which each species was affected.

**Figure 2 viruses-12-00605-f002:**
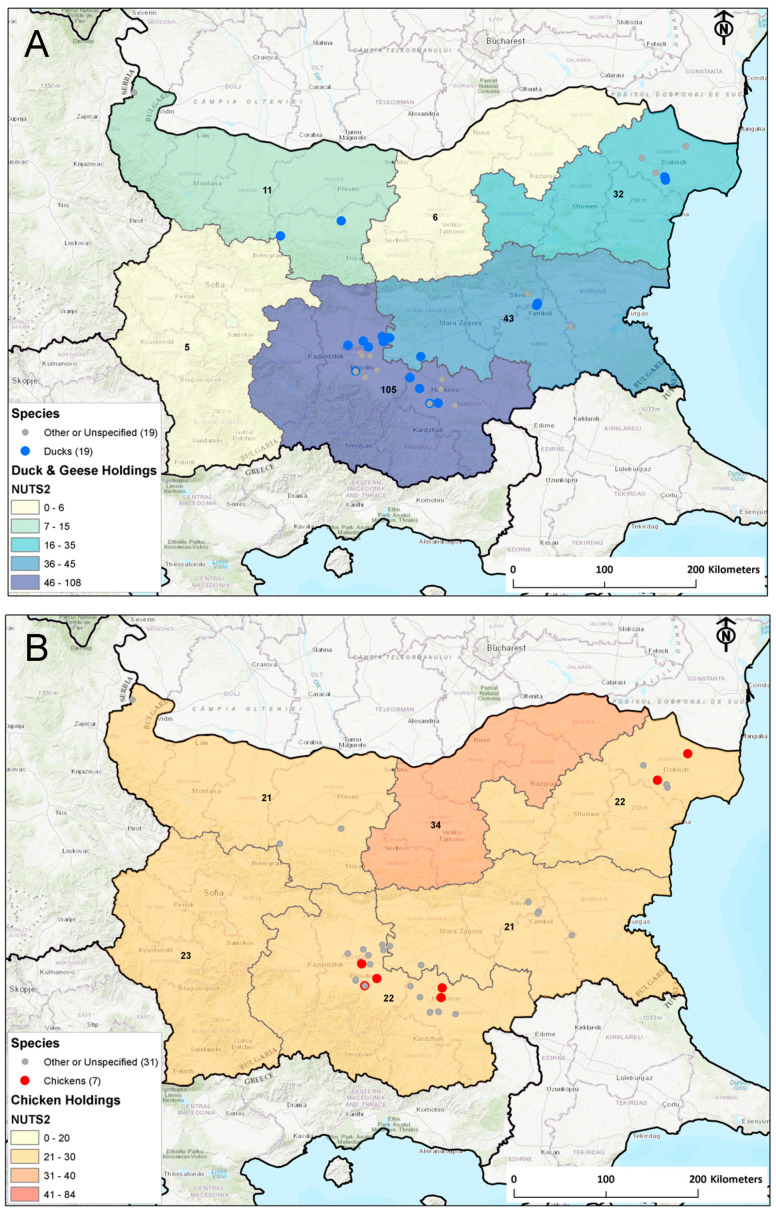
2017–2019 Poultry outbreaks in Bulgaria. Highlighted areas are NUTS2 (nomenclature of territorial units for statistics) areas with (**A**) labels indicating the number of ducks and geese registered holdings and (**B**) labels indicating number of chicken registered holdings. Grey dots refer to holdings with other poultry outbreaks.

**Figure 3 viruses-12-00605-f003:**
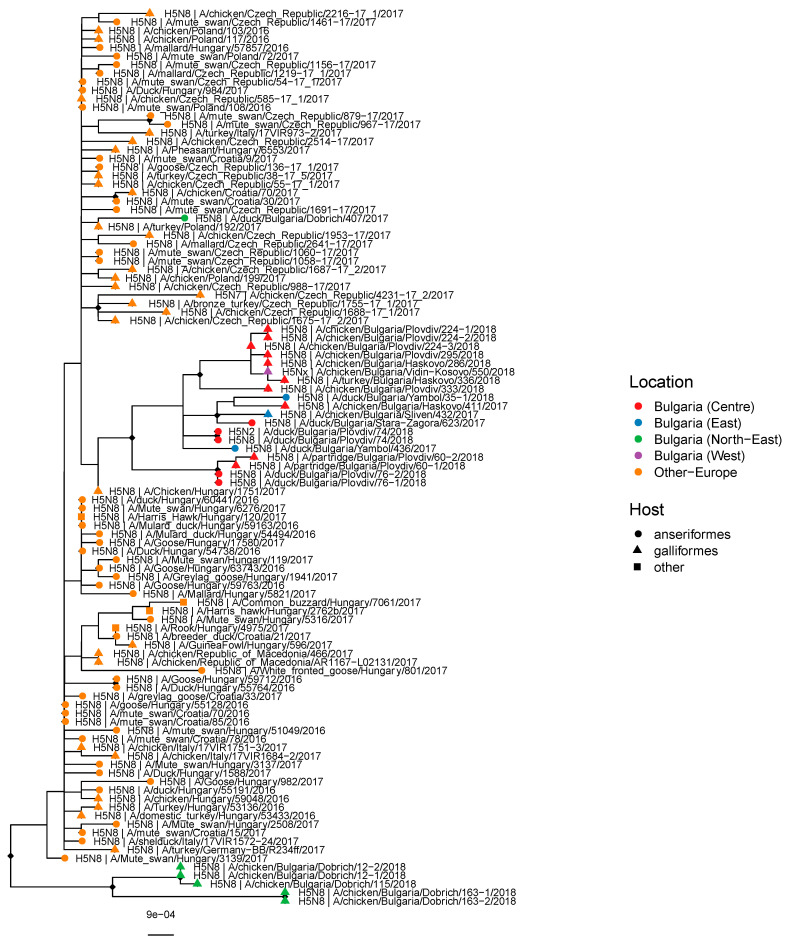
Mid-point rooted maximum-likelihood tree showing phylogenetic relationships between haemagglutinin (HA) sequences from viruses isolated from poultry outbreaks in Bulgaria (2017–18). Tip shapes indicate the host, while colors indicate location from which the virus was sampled. Diamond shapes at the nodes branch support values >90/100.

**Figure 4 viruses-12-00605-f004:**
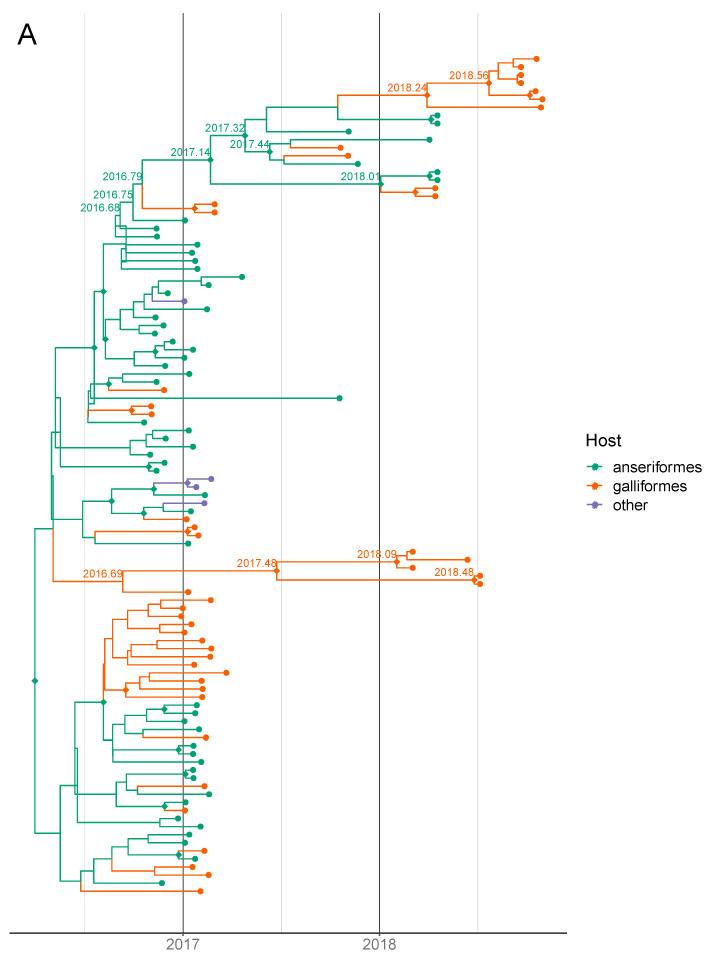
BEAST (Bayesian Evolutionary Analysis Sampling Trees) MCC (maximum clade credibility) tree showing reconstruction of host states at putative ancestral nodes of viruses isolated from poultry outbreaks in Bulgaria (2017–18, clusters highlighted gray). (**A**) Tip and branch colours indicate host states. Selected nodes of interest are labelled with putative times to the most recent common ancestor (TMRCAs). Diamond shapes at the nodes indicate posterior probability values >85/100. (**B**) Tip and branch colours indicate location states. Selected nodes of interest are labelled with putative location states. Diamond shapes at the nodes indicate posterior probability values >85/100.

**Figure 5 viruses-12-00605-f005:**
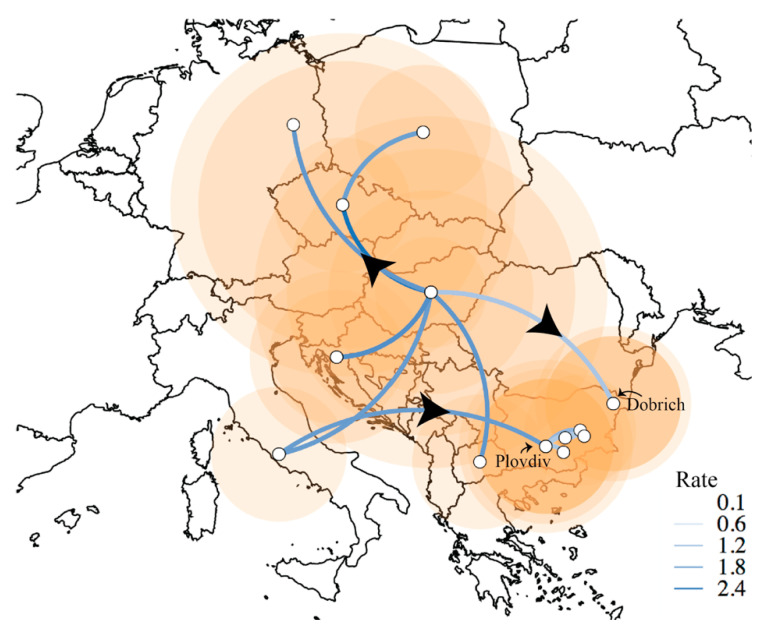
Summary of transmission rates between different locations as calculated from BEAST analysis using SpreaD3. Arrows show direction of transmission, the size of the orange circles corresponds to the cumulative number of cases.

**Figure 6 viruses-12-00605-f006:**
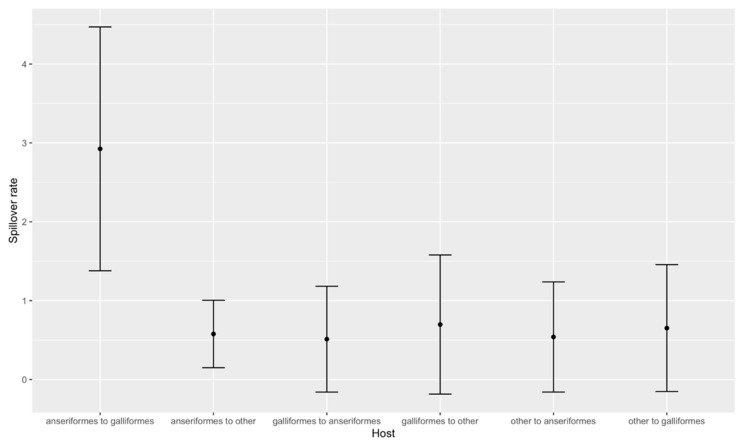
Summary of spill-over rates between different hosts (Anseriformes, Galliformes, other) as calculated from BEAST analysis. Error bars indicate 95% confidence intervals.

**Figure 7 viruses-12-00605-f007:**
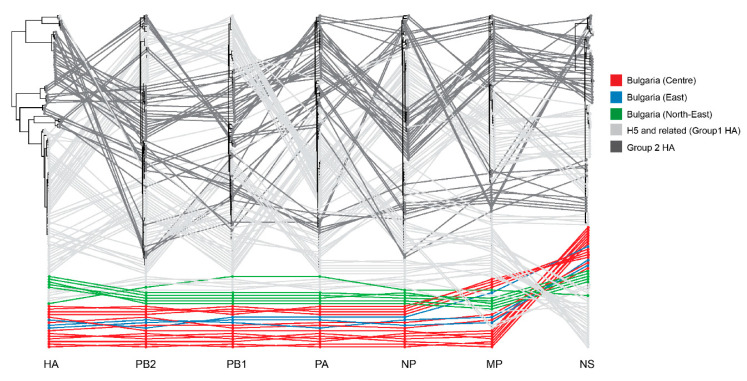
Visualization of phylogenetic incongruence as an indicator of differential ancestry of segment sequences i.e., reassortment. Maximum-likelihood trees for each segment are plotted with lines connecting the same virus strain across all trees. Colours indicate origin of viruses sampled in Bulgaria (red, blue, green, purple) during 2017–18 or related sequences outside of Bulgaria (grey).

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
