# Peer review of "Regional Transmission and Reassortment of 2.3.4.4b Highly Pathogenic Avian Influenza (HPAI) Viruses in Bulgarian Poultry 2017/18"

_viruses, 2020, doi:10.3390/v12060605_

Round 1

Reviewer 1 Report

The manuscript “viruses-801731” entitled “Regional transmission and reassortment of 2.3.4.4b highly pathogenic avian influenza (HPAI) viruses in Bulgarian poultry 2017/18” is a well-designed and well written research article. The authors have successfully used genomic and traditional epidemiological analyses to trace the origin and subsequent spread of H5 HPAI outbreaks within Bulgaria between 2017 and 2018. The data showed two separate introductions of the virus to Bulgaria, originating from different European 2.3.4.4b virus ancestors circulating in 2017, with minor reassortment events in matrix protein. The study also provided evidences for the contribution of domestic ducks to virus transmission into other galliform poultry.

Generally, the research presented in this article is of significant interest to scientists studying the flu virus. The data presented in this study are of notable scientific importance for veterinary authorities in Bulgaria. The English language of the manuscript is adequate; the quality of the figures and tables is satisfactory, the reference list cover the relevant literature adequately and in an objective manner.

Author Response

We thank the reviewer for their thoughts.

Reviewer 2 Report

This is a nice article about HPAI in Bulgaria in the Years 2017-2019.

The authors describe the epidemiology of the virus in commercial poultry during 2017-2019. influenza is known to have seasonal pattern, was this pattern observed? should the time frame presented be divided into two influenza seasons (2017-2018, 2018-2019).

The authors describe results of serological surveillance in commercial poultry without describing the scheme of surveillance in the materials and methods. I suggest the results of such surveillance will be summarized into a table including the season, production type, number of holding per type, number of sampled per type and number of positive per type. (ln 199-205)

Wild birds surveillance program was also not described in methods part.

In addition the authors describe results of surveillance in the neighboring countries as part of the results, it is not clear weather it is part of the current study or a comparison that should be addressed in the discussion. It seems out of the scope of the title or the article. it is of value to discuss. (ln 206-213)

Heading of table 1 states years 2017-2018 while the tables includes 2016-2018.

I suggest the authors will indicate year in each of figure 3 parts and present the percent of positive in each region. I would also divide the figure by the influenza season rather than by year.

The title of the article state years 17/18 as the subject presented yet the years presented changes from reports on 2016 to supplementary materials of 2020. I suggest the authors will refocus to the relevant period.

The main finding of the authors are the circulation of the virus within the commercial  poultry farms (ducks- ducks, ducks-others). the virus may originated from wild birds but the wide spread in commercial farms is due to low biosecurity. I suggest that conclusion will include the importance of maintain improved biosecurity measures rather than the risk of reintroduction from the wild. more over,the aim of the authors is the design better containment and preventive measures but this is not addressed.

ln 181 the word isolation is duplicated. I suggest rephrasing lines 186-187

Author Response

We thank the reviewer for their thoughts and comments which has helped improve this manuscript. Responses and details of changes made are indicated in italic text below.

This is a nice article about HPAI in Bulgaria in the Years 2017-2019.

The authors describe the epidemiology of the virus in commercial poultry during 2017-2019. influenza is known to have seasonal pattern, was this pattern observed? should the time frame presented be divided into two influenza seasons (2017-2018, 2018-2019).

We thank the reviewer for their comment. Avian influenza, in general, does not show seasonality in the same way that human influenza does. However, risk of incursion into poultry from wild birds is influenced by fluctuations in influenza virus infections in wild birds, which are driven by host ecological factors such as host population density and host annual life cycle behaviours such as migration, which could be seasonal.  The outbreak period may also be influenced by the stage of the sector specific production cycle i.e. duck sector production through the foie gras has some seasonal aspects being predominantly from spring to autumn.

However, once transmission into poultry has occurred, avian influenza is not seasonal in most instances. We have therefore not divided the timeframe by putative season – rather taken the timeframe as the epizootic or ‘outbreak’ period.

The authors describe results of serological surveillance in commercial poultry without describing the scheme of surveillance in the materials and methods. I suggest the results of such surveillance will be summarized into a table including the season, production type, number of holding per type, number of sampled per type and number of positive per type. (ln 199-205)

Wild birds surveillance program was also not described in methods part.

Surveillance within the EU is part of a broader programme of work involving both serological and virological surveillance. Data collected are the decision of the Member State, and in this case, we do not have all the variables recorded as the reviewer requests. In addition, there are often industry sensitivities surrounding the exact composition of an infected premise and thus these data would not be publicly available.

We note that we did not describe the wild bird surveillance programme in the methods because Bulgaria did not undertake such a programme during the study period. Methodology was not described as data was adapted fully from reference 41. There was no active wild bird surveillance in Bulgaria, and we have added a sentence to this effect in the manuscript. While we think this information is important context to our results, we agree that it might not best fit in that section. We have moved this section into discussion and have expanded on some details relating to the nature of this surveillance.

In addition the authors describe results of surveillance in the neighboring countries as part of the results, it is not clear weather it is part of the current study or a comparison that should be addressed in the discussion. it seems out of the scope of the title or the article. it is of value to discuss. (ln 206-213)

There was passive wild bird surveillance undertaken in all 3 countries. However, there were no detections in 2018, whilst historically H5N8 was detected in 2016 and 2017. This supports our assertion that poultry infections in BG in 2018 were unlikely to be due to direct infection from a migratory wild bird given that sensitivity of surveillance in wild birds in the region had previously been good. Since we did not undertake this surveillance as part of this study, we have moved this contextual information into the discussion.

Heading of table 1 states years 2017-2018 while the tables includes 2016-2018.

It should be 2016-2018, we have changed it accordingly.

I suggest the authors will indicate year in each of figure 3 parts and present the percent of positive in each region. I would also divide the figure by the influenza season rather than by year.

We have indicated the year in each part of the figure which is now in supplemental information.

We believe in this case that the positive proportion of samples is not a particularly helpful statistic. There was a single wild bird sampled in Plovdiv in 2017 that was positive. In the neighbouring region of Pazardzhik there were 9 positives from 14 samples. The map shows both the positive samples and the underlying number of samples taken and we think this is a more accurate representation of the surveillance effort.

As above, avian influenza in poultry has no clear seasonality. Thus, we present data for the outbreak study period.

The title of the article state years 17/18 as the subject presented yet the years presented changes from reports on 2016 to supplementary materials of 2020. I suggest the authors will refocus to the relevant period.

Whilst we are focused on poultry from 17/18, we are interested in the migratory season from autumn 2016 for wild birds as this would be a potential source of infection. The supplementary table S2 is the latest information from EMPRES-i, used to validate our assertion in the discussion that the poultry outbreak in Bulgaria continues beyond the study period for which we have sequence data available and are able to present analyses for in this manuscript.

The main finding of the authors are the circulation of the virus within the commercial poultry farms (ducks- ducks, ducks-others). the virus may originated from wild birds but the wide spread in commercial farms is due to low biosecurity. I suggest that conclusion will include the importance of maintain improved biosecurity measures rather than the risk of reintroduction from the wild. more over, the aim of the authors is the design better containment and preventive measures but this is not addressed.

We agree that the spread within the commercial sector is likely due to less-than-optimal biosecurity but we do not have direct epidemiological evidence for the method of spread in this case. It is likely multi-factorial – involving both breaks in biosecurity whether by indirect or direct means, and also complicated by the potential for cryptic infections in ducks, maintaining the virus and acting as a reservoir for further outbreaks. Therefore, we conclude that it is important to focus on improving biosecurity as this likely addresses the both the risk of introduction from wild birds and also the risk of spread within the poultry sector. According to suggestion, we have expanded on the mitigation and biosecurity measures in the discussion.

ln 181 the word isolation is duplicated. I suggest rephrasing lines 186-187

We have removed the duplicated word and rephrased those lines.

Reviewer 3 Report

The paper describes results of epidemiological surveillance of HPAI in Bulgarian poultry based on epidemiological information and genomic data.  The results are of high importance for HPAI control measures development. The authors repeatedly refer to lack of wild birds’ HPAI isolates sequences, that could make more clear the routes of local transmission of HPAI. Authors made comprehensive analysis of all available data. From the results, it is clear that HPAI was introduced in Bulgaria in the early 2017 and then was circulated in poultry population. The paper reports that subclinical infection in domestic ducks possibly leads to HPAI transmission between farms that makes obvious the need for monitoring ducks for HPAI even in abscense of any clinical symptoms in case of movement between farms.

The study is well designed and performed at high quality. The paper is clearly written and contains all necessary data supporting the authors’ conclusions.

Minor comments:

Figure 2 has incomplete legend: the figure capture formatting represents it as Fig 2A; 2A and 2B are not specified on the figure itself, it is not clear from picture what coloured and grey dots specify, needs text analysis. The addition of legend as in Fig 1 will make the picture more informative.

Line 181: misprint (“isolation” twice)

The funding information is in “Acknowledgments” section

Author Response

We thank the reviewer for their thoughts and comments. Responses and details of changes made are indicated in italic text below.

The paper describes results of epidemiological surveillance of HPAI in Bulgarian poultry based on epidemiological information and genomic data.  The results are of high importance for HPAI control measures development. The authors repeatedly refer to lack of wild birds’ HPAI isolates sequences, that could make more clear the routes of local transmission of HPAI. Authors made comprehensive analysis of all available data. From the results, it is clear that HPAI was introduced in Bulgaria in the early 2017 and then was circulated in poultry population. The paper reports that subclinical infection in domestic ducks possibly leads to HPAI transmission between farms that makes obvious the need for monitoring ducks for HPAI even in abscense of any clinical symptoms in case of movement between farms.

The study is well designed and performed at high quality. The paper is clearly written and contains all necessary data supporting the authors’ conclusions.

Minor comments:

Figure 2 has incomplete legend: the figure capture formatting represents it as Fig 2A; 2A and 2B are not specified on the figure itself, it is not clear from picture what coloured and grey dots specify, needs text analysis. The addition of legend as in Fig 1 will make the picture more informative.

We have described the meaning of the map colours in the legend including the coloured and the grey dots.

Line 181: misprint (“isolation” twice)

As above, we have deleted the duplicate word.

The funding information is in “Acknowledgments” section

We have added a separate heading for funding.

Round 2

Reviewer 2 Report

I find the revised version of the manuscript improved.